# Implementing DASH-Aligned Meals and Self-Measured Blood Pressure to Reduce Hypertension at Senior Centers: A RE-AIM Analysis

**DOI:** 10.3390/nu14224890

**Published:** 2022-11-18

**Authors:** Anuradha Hashemi-Arend, Kimberly S. Vasquez, Dozene Guishard, Moufdi Naji, Andrea Ronning, Glenis George-Alexander, Dacia Vasquez, Clewert Sylvester, William Pagano, Chamanara Khalida, Cameron Coffran, Teeto Ezeonu, Kadija Fofana, Dana Bielopolski, Roger Vaughan, Adam Qureshi, Jonathan N. Tobin, Rhonda G. Kost

**Affiliations:** 1Center for Clinical Translational Science, The Rockefeller University, New York, NY 10065, USA; 2Carter Burden Network, New York, NY 10021, USA; 3Clinical Directors Network (CDN), New York, NY 10018, USA

**Keywords:** DASH diet, hypertension, senior centers, older adults, aging, cardiovascular disease, implementation, RE-AIM, blood pressure, self-monitored blood pressure, intervention

## Abstract

Low-income, minority seniors face high rates of hypertension that increase cardiovascular risk. Senior centers offer services, including congregate meals, that can be a valuable platform to reach older adults in underserved communities. We implemented two evidence-based interventions not previously tested in this setting: DASH-aligned congregate meals and Self-Measured Blood Pressure (SMBP), to lower blood pressure (BP) at two senior centers serving low-income, racially diverse communities. The study enrolled congregate meal program participants, provided training and support for SMPB, and nutrition and BP education. DASH-aligned meals delivered 40% (lunch) or 70% (breakfast and lunch) of DASH requirements/day. Primary outcomes were change in BP, and BP control, at Month 1. Implementation data collected included client characteristics, menu fidelity, meal attendance, SMBP adherence, meal satisfaction, input from partner organizations and stakeholders, effort, and food costs. We used the RE-AIM framework to analyze implementation. Study Reach included 94 older, racially diverse participants reflecting neighborhood characteristics. Effectiveness: change in systolic BP at Month 1 trended towards significance (−4 mmHg, *p* = 0.07); change in SMBP reached significance at Month 6 (−6.9 mmHg, *p* = 0.004). We leveraged existing community-academic partnerships, leading to Adoption at both target sites. The COVID pandemic interrupted Implementation and Maintenance and may have attenuated BP effectiveness. DASH meals served were largely aligned with planned menus. Meal attendance remained consistent; meal satisfaction was high. Food costs increased by 10%. This RE-AIM analysis highlights the acceptability, feasibility, and fidelity of this DASH/SMBP health intervention to lower BP at senior centers. It encourages future research and offers important lessons for organizations delivering services to older adults and addressing cardiovascular risk among vulnerable populations.

## 1. Introduction

Cardiovascular Disease (CVD) is highly prevalent among the elderly. Over two thirds of those aged 60 to 79 and approximately 85% of those 80 and over have one or more forms of CVD [1]. CVD poses significant health risks for seniors, including stroke, myocardial infarction, heart failure, renal failure, and results in significant costs and suffering in terms of mortality, morbidity, disability, functional decline, and healthcare expenses [1]. Hypertension has been shown to increase individuals’ risk of CVD and stroke. About 1 in 3 US adults have high blood pressure, while only about 54% of those individuals have their high BP under control [2].

Racial and ethnic minorities have higher rates of CVD risk factors, including hypertension, and are more likely to experience heart attacks, strokes, and premature death. Hypertension control rates among communities of color often lag behind those of their white counterparts [2], with 44.6%, 45%, and 50.8% control rates among black, Hispanic and non-Hispanic white adults, respectively [3]. Lower socioeconomic status in the US has also been linked to a greater prevalence of CVD risk factors and higher incidence of and mortality due to CVD [2]. Thus, older adults from minority and lower socioeconomic backgrounds are at a particularly elevated risk of CVD.

Senior centers provide important resources for older adults, including a range of social, recreational, health, educational and nutrition programs, and other social services [4]. Around 10,000 senior centers across the US serve over a million older adults each day [5]. The NYC Department for the Aging (DFTA) funds meal services at 249 senior centers throughout the five boroughs serving adults over the age of 60. In the 2019 fiscal year, the agency provided 7.18 million congregate meals at senior centers, at no cost to 131,000 older adults [6]. Congregate meal programs at senior centers provide an opportunity for older adults to socialize and access low-cost nutrition services.

Low-cost health and nutrition services provided at senior centers may be particularly valuable for communities that face barriers to essential services due to racial and economic disparities.

The Carter Burden Network (CBN) operates four senior centers in diverse neighborhoods in NYC, providing subsidized congregate meals and other services. CBN operates a food pantry and typically provides nearly 300,000 congregate and home-delivered meals annually to food-insecure seniors aged 60 and over. In a 2015 survey among seniors in CBN’s East Harlem senior centers, 25% of participants reported cutting the size of their meals or skipping meals in the past 12 months for financial reasons, while only 5% stated they eat five or more servings of fruits and vegetables per day.

Since 2016, The Rockefeller University (RU) Center for Clinical and Translational Science, an NIH CTSA grantee, and Clinical Directors Network (CDN)—a primary care practice-based research network (PBRN) and AHRQ-designated Center of Excellence for practice-based research and learning—have engaged in a community-academic research partnership with CBN to identify and address unmet health needs among their clients. A RU/CDN/CBN 2016 pilot study at two CBN sites in East Harlem found that 83% of clients were overweight or obese and 84% had high, uncontrolled BP [7].

To address high BP and CVD risk observed among CBN clients, this community-academic research project implemented two evidence-based approaches to increase BP self-management by (1) improving nutritional composition of CBN meals using the Dietary Approaches to Stop Hypertension (DASH) diet, and (2) providing participants with personal devices and training for home Self-Measured Blood Pressure monitoring (SMBP).

The DASH diet is a plant-focused diet, rich in fruits, vegetables, and nuts, with supplementary inclusion of non-fat and low-fat dairy products, whole grains, poultry, fish, lean meats, and heart-healthy fats. It promotes a reduction in the consumption of red meat, sweets, and sugary beverages, and total and saturated fat and cholesterol [8]. The efficacy of the DASH diet has been demonstrated in two major studies backed by the National Institutes of Health (NIH), lowering BP in as little as 14 days [9,10]. However, very few studies have reported on adapting the DASH diet to a community setting [11]; some have focused on education about the diet [12] and none actually changed the meals provided at senior centers. To our knowledge, the DASH diet has never been studied in a congregate meal setting among community-living, food-insecure seniors.

The research shows that Self-Measured Blood Pressure (SMBP) monitoring improves blood pressure control [13,14,15,16,17]. Research on SMBP with older adults shows that, while seniors can benefit from self-monitoring, complex technology such as blue-tooth wireless connectivity may present challenges [18]. One senior center-based BP monitoring program uses peer volunteers [19] and a recent trial of a digital SMBP intervention found sustained engagement among older adult participants, with substantial minority representation [20]. While these lifestyle interventions have proven effectiveness in clinical settings, more information is needed about their translation into implementation strategies at senior centers, where many vulnerable older adults seek services [21,22].

Here, we use the RE-AIM framework, one of the most commonly applied frameworks in implementation research, to report on program implementation of the DASH + SMBP interventions. The RE-AIM framework expands the scope of reporting studies beyond program efficacy and effectiveness, to consider the intervention’s context and conditions and improve understanding about generalizability [23]. The framework was developed to facilitate translation of research to impact community and population health [23]. Organizing outcomes according to the five RE-AIM elements—Reach, Effectiveness, Adoption, Implementation, and Maintenance—allows direct examination of the feasibility, fidelity, acceptability, and challenges and leverage lessons learned for scale-up and adoption of effective multi-component interventions into other settings, thus increasing both the external validity and public health impact.

## 2. Materials and Methods

### 2.1. Research Ethics

The Rockefeller University Institutional Review Board (IRB) approved the project for all sites under a Single IRB mechanism. The trial was registered at clinicaltrials.gov (NCT03993808). Informed consent was obtained before initiating study activities.

### 2.2. Study Design

We designed an open-label study implementing (1) DASH-aligned menus served by an ongoing congregate meal program, and (2) educational and behavioral support for home SMBP monitoring among lower income community-living seniors attending two senior centers in New York City.

### 2.3. Study Aims

The study tested whether the combined interventions lowered systolic blood pressure at Month 1 (primary aim) and Months 3 and 6 (secondary aims) [24]. Increasing the fraction of individuals with “controlled BP” was a co-primary aim. BP was measured by an independent healthcare contractor, Vital Care Services https://www.myvics.com/ (accessed on 1 September 2021), using the Fora P20b BP Monitoring System according to their Community Telehealth Standard Operating Procedures, incorporating support from a Telehealth Technician Associate (TTA) on-site, and from a registered nurse remotely [24,25]. Additional secondary aims addressed program implementation to: (1) Implement DASH-concordant meals by adapting CBN’s existing DFTA-approved menus with input from the CBN clients (by engaging clients in townhalls and taste tests) to optimize client acceptance of the intervention; and (2) Provide up to two DASH-aligned meals/day to CBN clients; (3) Enhance the value of nutritional service programs by reducing waste and maximizing cost efficiency; (4) Support cognitive and behavioral change by providing cardiovascular risk reduction education, self-measured blood pressure (SMBP) training, nutritional and cooking education, health education, and medication adherence training; (5) Provide positive feedback and enhance self-efficacy through on-site and home SMBP monitoring; and (6) Leverage and grow a sustainable, multi-stakeholder partnership. This partnership included an academic center (Rockefeller University), community senior services (Carter Burden Network), and community health center organizations; a Practice-Based Research Network (PBRN—Clinical Directors Network, CDN); government agencies; and community-living seniors to incorporate client-centered, culturally responsive nutritional preferences and priorities into the design, conduct, implementation, analysis and dissemination of the intervention program.

### 2.4. Study Participants and Settings

The study was initially designed to enroll 200 adults, 100 at each of two CBN sites. Eligible clients were age 60 or older, consumed at least 4 congregate meals/week at the study sites, and provided informed consent. There was no BP threshold for entry since all clients were eligible to receive their meals at CBN. Site 1, located in East Harlem, serves breakfast and lunch seven days a week; in addition, 125–150 clients attend one or more congregate meals each week. Social determinants of health among the neighborhood population include high levels of poverty, food insecurity, and low rates of higher education [26]; many clients at this site are racial and ethnic minorities and live alone; furthermore, 13% do not speak English. Site 2, located on the Upper East Side of New York City, serves lunch five days a week to about 120 clients. The majority in the neighborhood are White and hold a high school diploma [26]; many clients at Site 2 are economically distressed, live alone in rent-stabilized apartments, and some report homelessness (CBN data, unpublished).

### 2.5. Study Interventions

For the nutritional intervention, RU dietitians and CBN food service staff worked in close collaboration with the dietitians at the New York City Department for the Aging (DFTA) to adapt the existing 6-week congregate meal menus to align with both DFTA standards and DASH principles. These DASH-aligned, DFTA-approved menus replaced all congregate meals served at the two sites for a six-month period. The educational programs were provided in the days and weeks immediately following initiation of the DASH-aligned meals to enhance knowledge about nutrition, blood pressure management, and medication adherence, and included bilingual presentations on CVD risk and managing high blood pressure by a CDN primary care physician (WP).

Participants each received a personal blue-tooth enabled home blood pressure monitor (10 Series, OMRON Healthcare, Kyoto, Japan) and training in its use and were instructed to begin SMBP at home as soon as the device was provided to become familiar with the practice prior to the nutritional intervention. Participants received a SMBP diary card and were instructed to record their measurements and share their diaries with their primary care clinicians to help manage medications and monitoring.

### 2.6. Study Measures

Concordance of the DASH-aligned congregate meals as-served, with the meals as-planned (fidelity), was measured by tabulating the total servings of each of the DASH components provided and comparing it with the planned menus meeting DASH recommendations. Total servings below the planned threshold were rated unacceptable; those exceeding the planned threshold were rated as unacceptable (e.g., fat allowance) or acceptable (e.g., vegetable allowance) according to DASH guidelines. Primary Outcome **measure** was the change in systolic blood pressure as measured onsite by an independent healthcare contractor (Vital Care), according to written standard operating procedures, at Month 1 of the intervention compared to Baseline. A co-primary outcome measure was the change in the percentage of participants with blood pressure considered “controlled”, (<140/90 mmHg for adults age <60 years and <150/80 mmHg for adults age >60 years) as defined by the Eighth Joint National Committee (JNC-8) [25]. Primary blood pressure outcomes were previously reported [24].

Physiologic and survey measures were assessed at Baseline (prior to initiation of any study interventions) and at Month 1, Month 3, and Month 6. In addition to BP, physiologic measures included pulse and BMI. Self-administered surveys included 12 validated instruments addressing demographic, psychosocial, food security, nutritional, and health-related questions [24]. Most of the survey instruments were validated for use in low-literacy populations in both English and Spanish.

Meal attendance was tracked through the DFTA-sponsored system, “Peer Place”, whereby seniors signed-in electronically using a swipe tag at each meal.

Meal Satisfaction was measured >2 times weekly before and throughout the nutritional intervention. Colorful graphic “Smiley Likert Food Satisfaction” rating cards in English and Spanish were developed and distributed during meals to collect anonymous ratings and open-text comments. In alignment with CBN culture and dining room logistics, all clients (enrolled and unenrolled) could provide feedback on meal/menu acceptance. Initial plans to distinguish between feedback from enrolled/not enrolled seniors were not feasible with dining room flow.

Plate Waste was assessed for a non-sequential convenience sample of 20 plates, two days each week at each site, on the days when meal satisfaction was assessed to assess client preferences for specific menu items, inform recipe acceptance and change, and minimize waste. We used the Quarter Waste method [27], a reproducible and reliable method that uses visual assessment by an experience observer, to determine the fraction of each DASH food group left on the plate (0%, 25% 50%, 75%, or 100%). We designed a custom data capture system, using REDCap [28] to integrate daily menu information into the Quarter Waste data collection forms, making the plate scoring tool specific and user-friendly.

SMBP data were downloaded by study staff via Bluetooth into an Excel file containing participant ID, date, time, SBP and DBP, and heart rate. Participants met with a team member every 2–4 weeks to download the data stored on their personal SMBP device. During these interactions, the study team encouraged participants to take at least one daily SMBP reading, reinforced key concepts, addressed any questions or concerns, and provided advice to resolve common barriers. Qualitative data on barriers and facilitators of SMPB were collected as interaction notes. From downloaded SMBP data, we calculated the mean systolic BP for each individual, for each week of study and compared mean change from baseline across the study timeline.

To structure our analysis of the project implementation, we selected the RE-AIM framework. In Table 1, we provide the RE-AIM dimensions, their definitions, and align the appropriate study data for analysis using this framework and the data sources.

## 3. Results

This section organizes study findings and the related discussion by the 5 RE-AIM dimensions defined above in Table 1.

### 3.1. Reach: The Representativeness of Individuals Willing to Participate in Program, and Reasons Why or Why Not

#### Study Enrollment and Participant Demographics

The study screened 111 CBN clients and enrolled 94 (47% of target enrollment) adults aged 60 years or older, who self-reported eating at least four congregate meals per week at one of the two CBN sites. Ninety-six participants did not participate in screening either because they did not speak one of the study languages (validated instruments were not available in Chinese), or they were not interested due to the time-consuming nature of study activities. The CONSORT diagram, Figure 1, previously reported [24], outlines the flow of participants through each stage of the study and reasons for declining to participate.

Of the older adults who declined to enroll, most cited competing priorities and the time-consuming nature of the study or were ineligible because they reported eating less than four meals each week at the sites. Some had cognitive impairment, were unable to attend baseline and Month 1 visits, or were planning to discontinue attendance at congregate meals. Among the 94 who enrolled, 10 were persistently unavailable for the Baseline study visit and withdrew before the meal intervention began. Eighty-four participants stayed in the study to complete Baseline assessments.

The demographic characteristics of study participants were comparable to the clients overall at the two CBN sites who attended congregate meals during the study period, with two exceptions (Table 2). More Persons of Color were represented in the study cohort (44%) than in the overall congregate meal population (33%), largely due to higher enrollment of Black vs. White clients (30% vs. 10%). Asian participants were significantly underrepresented in the study group (2% vs. 20%). This was expected as the validated survey tools used in the study were not available in validated Chinese translations; therefore, clients who spoke only Chinese could not be enrolled in the study. Nutrition and BP educational materials were made available in Chinese as well as English and Spanish as a benefit to all CBN clients. Female participants were somewhat overrepresented in the study; age distribution was generally comparable.

The prevalence of hypertension among CBN clients who elected not to participate in the study is unknown. However, unpublished data about Site 1 clients collected during a 2015 pilot study suggest the current cohort had a similar distribution of hypertension as the overall CBN client population. The prevalence of Stages 2–3 of hypertension was similar in the DASH study cohort and 2015 pilot populations (44% vs.45%, respectively). There were more participants with Stage 1 hypertension in the DASH versus pilot population (31% vs. 19%, respectively) and fewer individuals with normal or minimally elevated BP in DASH vs. the pilot (23% vs. 35%, respectively). Thus, the study participants are generally representative of the overall population served by CBN.

### 3.2. Effectiveness

#### 3.2.1. Change in BP

The primary outcomes have been previously reported [24]. Briefly, sixty-one participants completed both the Baseline and Month 1 measures necessary to be included in the primary outcome analysis. Participants experienced a mean decrease of 4.41 mmHg (*p* = 0.0713) in their SBP at 1-month of DASH intervention compared to Baseline. A logistic regression analysis revealed that age and higher baseline BP were significantly associated with decrease in BP (*p* = 0.04 and *p* < 0.001, respectively), and higher BMI was associated with an increase in BP (*p* = 0.02). In the secondary outcome analysis of BP change over time, among the 60 (71%) participants who continued SMBP at home through Month 5 or 6, there was a mean decrease in systolic BP of 6.9 mmHg (*p* = 0.003) [24].

#### 3.2.2. Change in Psychosocial and Health Behavior Measures/Surveys

Study surveys included psychosocial and behavioral measures including medication adherence, food behavior, and medical care. Most measures were insensitive to change over this six-month study. There was a small increase in the number of Site 1 participants who reported having a doctor or healthcare provider at Baseline (91%) to Month 6 (97%). In contrast, the number of Site 2 participants reporting having a doctor or healthcare provider at Baseline (92%) was decreased at Month 6 (74%). This comparison may be confounded by the fact that Month 6 for Site 1 was just at the onset of the pandemic (March 2020), whereas Month 6 for Site 2 fell several months later into the early months of the COVID pandemic in New York, which limited patients’ ability to seek medical care (Figure 1). The mean number of hospitalizations and ER/urgent care visits was low (<1) at baseline for both sites and did not change across time points.

The study lacked measures to track DASH adherence outside of the congregate meal setting. We designed an exploratory short survey to assess eating habits according to DASH parameters, but it performed inconsistently and will need further validation.

### 3.3. Adoption

#### 3.3.1. CBN, RU, and CDN Partnership

The existing CBN, RU, and CDN community-academic partnership was instrumental in project adoption at two CBN senior centers. The community organization (CBN) was the primary recipient of the federal grant supporting the project, and the academic partners (CDN and RU) were sub awardees. The project team (including CBN, RU, CDN, and Vital Care Services partners) met monthly over the 24 months of the project, alternating between locations at CBN and RU offices. These meetings ensured alignment on the various aspects of the project by enabling members to provide updates on their responsibilities and share challenges that may result in timeline or procedural adjustments.

#### 3.3.2. External Advisors

Program adoption also relied on engagement with a group of diverse external partners and advisors. A DASH Project Advisory Board was convened at the outset of the project and met quarterly during implementation and analysis to review progress, problem-solve, and provide considerations for broader adoption. The Advisors were leaders in other senior-service, health, government, and nutritional service organizations serving older adults in New York City (Appendix A). Of the external advisors, four (33%) were affiliated with government agencies, seven (58%) with non-profit organizations, and one with a public hospital (8%). The results, outcomes, limitations, and conclusion of the study were shared with the Advisory Committee at the end of the study. They provided their perspectives on the feasibility and generalizability of broader adoption.

#### 3.3.3. DFTA

The New York City government agency that subsidizes the congregate meal program, DFTA, was a key collaborator and provided consultation in menu development. DFTA’s approval was required before any changes to the menus could be implemented at the sites. Nutrition programs affiliated with DFTA use a web-based menu application called Simple Servings to build their menus and submit them for approval [30]. Development of the DASH-aligned menus and recipes required iterative revision due to incongruities between DASH standards and the restrictions of the Simple Servings Menu platform. Simple Servings relies on United States Department of Agriculture (USDA) guidelines and definitions, which differ from the DASH rubric. Through analysis of USDA and Simple Servings starch classifications, trial and error, and ongoing negotiations with the DFTA Supervising Nutritionist, the team ultimately achieved approval for a different six-week cycle of menus for each of the sites with recipes that satisfied both rubrics.

#### 3.3.4. Collaborative Study Workgroups

Smaller workgroups enabled the project team to allocate responsibilities to team members based on role and expertise. The workgroups met regularly to address their respective project responsibilities and shared progress and received guidance from the larger group during monthly project team meetings.

The Nutrition Workgroup included bio-nutrition leadership and staff from RU, the RU PI and project manager, food service leadership from CBN, and a senior dietician from DFTA to collaborate on menu and recipe design, DASH alignment, accommodation of food preferences, meal implementation, and achieving approval. The Nutrition Workgroup also developed and tested the Plate Waste assessment approach and conducted plate waste assessments during the study. The nutrition team worked with the IT/Data workgroup to design data collection tools for plate waste activities.

The IT/Data management group, with the PIs and project manager, managed configuration of electronic tablets for survey collection at the sites using REDCap based surveys and syncing with cloud-based database, providing Wi-Fi hotspots. The IT/Data management group team created data collection tools for the plate waste activities and an overall project database. IT assisted in configuring the SMBP devices for downloading and transferring to the RU database. The data management staff assisted in transforming SMBP downloads, and DFTA meal attendance data, so the data could be analyzed.

### 3.4. Implementation

#### 3.4.1. Congregate Meal Fidelity: Delivery of Meals As-Served versus As-Planned

The DASH-aligned menus were implemented at Site 1 on 15 October 2019 (for breakfast and lunch) and at Site 2 (lunch) on 3rd February 2020. The six-week DASH-aligned menu cycles continued at both sites until COVID closures in late March, 2020. To track alignment of meals as-served with DASH-aligned meals as-planned, each food component served was tallied and tracked (Table 3). Site 1 was delivering two separate DASH-aligned menus (breakfast and lunch), as opposed to only one meal at Site 2 (lunch) and had twice as many meals and requirements per food group to follow. By Week 3, meals as-served were highly aligned with the DASH meals as-planned, aside from 1–2 servings of fruits and grains weekly. At Site 2, during week 1, clients were offered a choice between two fruits and received one fruit serving rather than the planned two. After the 1st week, however, alignment with planned DASH servings improved.

#### 3.4.2. COVID Interruption/End of DASH Congregate Meals (Feasibility)

Due to the onset of the COVID-19 pandemic, all senior centers in New York City were mandated to suspend in-person services starting 16 March 2020. As a result, all DASH-aligned congregate meal services were suspended from that date. In the initial weeks after this suspension of services, CBN continued to provide DASH-aligned “grab-and-go” meals, which were largely aligned with the project menus, minus one serving of vegetables that did not fit into preformed delivery trays, which had a fixed array and number of compartments. Later in March, DFTA implemented a comprehensive city-wide home-delivered meal program for congregate meal recipients, Get Food NYC, and requested that senior centers stop providing grab-and-go meals. While CBN supported the Get Food NYC program by receiving bulk meals, coordinating with clients, and registering new participants, CBN could no longer ensure DASH alignment of meals. The Bionutrition workgroup evaluated the grab-and-go meals initially provided by one vendor and found they were largely aligned with DASH. However, when the Get Food NYC program switched to a second vendor in May 2020, the meals were no longer DASH-aligned. We marked the end of the grab-and-go services as the end of our ability to reliably provide the DASH-aligned meal component of the intervention.

#### 3.4.3. Client Engagement in Menu Development

A six-week menu cycle was developed for each of the two sites, adapting existing menus according to the requirements of the DASH diet as well as site-specific client preferences. All CBN clients received the same DASH-aligned meals. The logistics and costs of asking kitchen staff to prepare and manage two separate menus simultaneously was not feasible. Participants consuming both breakfast and lunch at study sites would be provided 70% of DASH daily requirements for protein, grains, dairy, vegetables, fruits, fats, and 70% of weekly requirements for seeds/legumes, and not exceeding weekly limits for sweets; participants only consuming congregate lunches would be provided 40% of the same DASH components. Details about changes made to the congregate meal menus are provided in Appendix B and Appendix C.

The preferences of the senior center clients also informed the menu changes. Prior to DASH menu development, the Site 1 Director had conducted a DFTA food survey to assess preferences among clients. In that survey, clients requested that more menu items including chicken, fish, and vegetables. These preferences aligned with the DASH diet and provided encouragement that the planned menu changes would be desirable to CBN clients. During menu planning, Town Halls were organized at both sites with CBN clients to provide tastings in order to gauge feedback on the menu changes and to prepare the seniors for the menu changes prior to DASH-diet implementation. Tasting samples of two proposed menu items were distributed during Town Hall Meetings (a vegetable only and a vegetable/grain preparation). While a few seniors expressed some reservations about the menu changes, most indicated overwhelmingly positive acceptance of the new items (Table 4).

#### 3.4.4. DASH Congregate Meals Attendance and Satisfaction (Client Acceptability)

DASH-aligned congregate meals were implemented at the site level so that participant and non-participant clients were provided with the same DASH-aligned meals. Only study participants’ meal attendance was tracked. Attendance at both sites 30 days prior to DASH implementation was 3.76 ± 0.81 compared to 3.63 ± 0.66 in the month following DASH (*p* = 0.27). Attendance at Site 1 was 4.15 ± 0.35 during the first month of DASH implementation, dipped slightly around Month 3 (3.76 ± 1.06), but increased again to 4.13 ± 0.79 until Month 5 (*p* = 0.88). However, meal attendance remained consistent after introduction of the DASH-aligned menus compared to participants’ attendance immediately prior to implementation and demonstrated the acceptability of DASH-aligned congregate meals among senior center clients.

Overall, there was widespread acceptance of the changes to DASH aligned menus. The Smiley Likert cards use low literacy smile icons on a 5-point visual (Likert) rating scale to ask respondents to rate their meal. The cards proved popular and provided important feedback (Figure 2). In the first week of the intervention at Site 1, food acceptance declined somewhat. The team looked at participant comments to identify reasons behind the dip in satisfaction and worked with the food service staff to further refine the menu changes. Within a week, the change was implemented, and the acceptance scores rose. Over time, the percentage of the scores in the highest rating category were higher during the DASH implementation than it had been in the weeks prior to starting DASH meals. Initial verbal complaints that “vegetables do not belong on the breakfast menu” were overcome by rising satisfaction scores.

#### 3.4.5. Plate Waste (Acceptability)

Analysis of Plate Waste (Figure 3) revealed minimal waste of fruits, most vegetables, dairy, seeds, and oils. Certain grain options, such as bulgur, quinoa, brown rice, were discarded in moderate amounts throughout the observation period. Other grains (whole grain bread, whole wheat pasta) were consistently consumed. These findings are consistent with verbal feedback that brown rice and other whole grains were less palatable. A major limitation of the Plate Waste analysis was made apparent only after the study was underway. Some wasted portions appeared as large or larger than the intended serving size. Portion control for serving size was specified in the menu plans but was not monitored. We speculate that some items in plentiful supply, such as rice and grains, may have been served overgenerously, making plate waste a qualitative measure, but not a reliable quantitative signal of participant preference for certain food options.

#### 3.4.6. Food Cost (Feasibility)

Implementation of the DASH-aligned menus included ingredients that were less pre-processed and required more labor and space for meal preparation. DASH meals also required more servings of fresh fruit and vegetables than most American diets. Our initial plan—to provide the full DASH daily recommended fruit and vegetable servings in two meals—was projected to increase the cost of the congregate meals by almost 30%. This approach also would have added too many calories in the space of two meals to meet the Simple Servings guidelines based on USDA recommendations and required by DFTA. Despite concerns that some clients might be subsisting on two rather than three meals a day, the menu framework could not accommodate exceptions. Therefore, the team revised the menus to reduce the added costs, provide fewer servings (less than 100%), and reduce the total calories. With the revised menus, comparison of food costs during the DASH intervention period ($248,885.78), with costs of the same period in the preceding year ($224,920.73), revealed a 10% increase. CBN sought and received assistance from the Hearst Foundation to support the addition food costs through the study period.

#### 3.4.7. Home Self-Measured Blood Pressure (SMBP) Implementation (Fidelity and Feasibility)

Ninety-eight percent of Site 1 and 97% of Site 2 participants received in-person training on using their personal BP devices. Some Site 1 participants did not attend training or begin self-monitoring until after the DASH intervention had begun. All Site 2 participants received training and home monitors by Day 1 of the DASH intervention. Training was reiterated biweekly at data download visits with study staff. In-person meetings to assist participants with data downloads were interrupted by COVID, requiring the team to switch to remote support. This change was challenging as participants required significant hands-on support for managing the blue-tooth enabled data downloads. The team adjusted by developing detailed SOPs for providing remote support and regularly checking in with participants.

There were several technical challenges with the Omron BP devices. Blue-tooth enabled devices are configured for personal data download to a smartphone and sharing with a provider via email. The devices are not designed to be used in close proximity to other similar devices, and it can be challenging to keep downloads distinct in the absence of personal identifiers on the files). Multiple technical work-arounds were needed to keep data transfers distinct when syncing and downloading in proximity at the senior center. Data transfer did not work if the memory was too full, if the BP monitor batteries were inadvertently removed by the participant while the monitor was turned on, or if the monitor became unpaired (Bluetooth) to the tablet. After COVID shut-downs prevented in-person meetings, support for SMBP was provided remotely. This introduced another set of challenges due to limitations in seniors’ internet access, the availability of appropriate devices, and familiarity and aptitude with technology. Participants were given the option of reading out their BP measurements from their monitor or BP diaries over the phone in lieu of Bluetooth enabled data transfer. This was effective but was time consuming, may be subject to selective reporting of BP readings (e.g., omitting high values), and would be difficult for senior centers to scale. Because of participants’ unfamiliarity with the technology, we infer that few participants kept data downloads for themselves and did not share their SMBP data with their clinicians.

Seventy-six (90%) of participants used their SMBP device and 47/84 (56%) completed SMBP measures at least once a day through Month 1. Twenty-five (30%) continued taking measure through Months 5 or 6 [24]. A comparison of participants who stopped SMBP after Month 1 versus those who continued showed a higher proportion of black or female participants among those who continued SMBP to the end of the study [24].

During 321 visits with study staff to download SMBP data, participants reported barriers and facilitators to SMBP monitoring that were logged in interaction notes (Table 5). Participants reported that regular reminders from spouses, housemates, calendars, and notes helped them remember to measure SMPB regularly. Participants cited forgetting, emotional stresses, and confusion about study and technical procedures as barriers to taking regular SMBP measurements.

#### 3.4.8. Health Education Program

The study presented health education in English and Spanish on (1) blood pressure management and medication adherence and (2) nutrition. The educational session on BP management and medication adherence was presented in person by an experienced family practice physician. Simultaneous translation was provided in Spanish. The session was provided multiple times at each site to maximize attendance; 29/45 (64%) of Site 1 participants and 38/39 (98%) of Site 2 participants attended educational sessions. Health education on nutrition was provided in classes. The first class, “DASH for Overall Good Health”, was delivered in person as planned on two different dates at both sites. Thirty-three of 45 (73%) of Site 1 and 35 of 39 (90%) of Site 2 participants attended one or both sessions. After COVID-related closures, the second class (“Nutrition Facts Labels 101”) was delivered virtually unsuccessfully. At Site 1, attendance was very low (n = 4) because most seniors were unfamiliar or unable to access the Zoom technology early in the pandemic. Since the meal intervention and assessments were concluding, nutritional training was not continued for Site 2 participants.

### 3.5. Maintenance

This six-month program was not designed to assess long-term client or organizational impact. For CBN, this was their first federal research grant, reflecting the organization’s increasing capacity to manage inter-agency projects and offer comprehensive cross-sector response to complex challenges exhibited by their clients. After undertaking this project, CBN subsequently partnered with Public Health Solutions (the largest public health nonprofit serving New York City) on another three-year Nutrition Innovation grant from ACL, focusing on community and partnership building in East Harlem to improve nutritional and health outcomes for seniors residing in public housing. CBN was not able to resume DASH menus after the pandemic in the absence of additional resources to support increased food costs for DASH. We are working on securing funding to test models of sustainability.

## 4. Discussion

We described here a preliminary study that sought to implement two evidence-based interventions (DASH diet, SMBP) in the setting of senior centers with congregate meal programs. Using the RE-AIM framework, we have reviewed the evidence of successful implementation and effectiveness, in spite of significant COVID-19 related challenges.

### 4.1. Reach

Overall, the reach of the study was largely representative, signaling the program’s acceptability among the target population and the feasibility of enrolling participants in this type of intervention. Some senior center clients were not eligible for participation as they did not speak one of the study languages or declined to participate due to the time-consuming nature of study activities. In the future, when the program is replicated without validated surveys and the burden of study activities is reduced, we anticipate greater participation rates. The level of HTN among the cohort was similar to that of the prior HTN study population though overall the group had significant numbers of individuals with normal, minimally elevated, or Stage 1 HTN, whereas the DASH diet’s effectiveness might be more dramatic in a group with higher Baseline BP. This limitation, and the slight skewing of the study cohort toward female clients, with fewer of the 85+ population, could be addressed with more stringent eligibility criteria though in this study it might have further compromised accrual.

A specific study limitation in achieving optimal reach was the lack of validated instruments and services in Chinese to target and enroll Chinese-only speakers, leading to a lower enrollment of Asian participants. Ultimately, study surveys were not essential (and burdensome to study participants and staff), thus future iterations of the study could omit them and hire bilingual English/Chinese-speaking staff to enroll and instruct participants who only speak Chinese.

### 4.2. Effectiveness

The DASH intervention, which provided 40–70% DASH components on the several days of the week that clients consumed them (mean 3.7 meals/week), was meant to model the flavors and components of DASH and enhance skills and self-efficacy in BP self-monitoring. The significant lowering of BP detected in the Month 5/6 measures (despite the interruptions of COVID-19) in those participants who continued the intervention suggests that this combined intervention can be effective in individuals who adhere to the program. In future studies, survey collection should be minimized, and anticipated data well justified to reduce the burden on the study team and participants. DASH adherence outside the congregate meal setting should also be tracked (e.g., via food diaries, rapid food frequency questionnaires, cellphone meal photographs) to assess the impact of the nutritional intervention on behavioral change.

### 4.3. Adoption

A history of cooperation and collaboration within the multi-institutional project team, and engagement of external advisors laid the foundation for project adoption at the two study sites. Partnering with DFTA was essential to program fidelity since the DASH-aligned menus required DFTA approval before CBN could implement them. The working groups ensured that project partners were aligned and working together throughout study design and implementation. This groundwork, which enabled program implementation at the two targeted sites, will also help sustain the nutrition and behavior program and its blood pressure lowering impact and facilitate broader adoption by additional senior centers in the future.

### 4.4. Implementation

The study engaged with staff at two senior centers and their clients to ensure that menu adaptations aligned with both the dietary guidelines and client preferences through townhalls during menu planning, and during implementation, through tracking meal satisfaction and adjusting the menu according to findings. Implementation was also affected by institutional restrictions—DASH aligned meals had to correspond with the funding agency’s nutritional regulatory requirements. Meal satisfaction was high, and meal attendance was essentially unchanged after introduction of the DASH-aligned menus, demonstrating the acceptability of the DASH diet and the feasibility at the client, organization, and regulatory agency levels of implementing DASH aligned meals at senior centers.

All CBN clients, enrolled and unenrolled, provided the same DASH-aligned meals for program logistical reasons. It was not possible to distinguish study participants from general CBN clients in the dining room for privacy reasons. This complicated plate waste assessments as we were unable to distinguish among the plates from diners who were study participants and had received nutrition and BP education, SMBP training, and encouragement, and might therefore be more motivated to consume the altered menus, versus diners who were eating the DASH meal without any of the supporting information. Additionally, since we were not able to confirm whether serving size was well controlled, especially for grains, we are unable to draw quantitative conclusions from the observations of leftover foods. We are confident that the complete consumption (little waste) for fruits and most vegetables is an encouraging sign given their importance to DASH. Discarded food presented an opportunity to reduce waste by (1) reviewing portion control and (2) attending to food waste patterns and altering recipes of food items that are consistently discarded. Plate waste assessments are most valuable when conducted complementary to periodic meal satisfaction assessments, to identify more and less popular items and make targeted adjustments. The initial project projected providing 100% of DASH components in meals which would have incurred a food cost increase of 30%. Dietary limits on calories forced the dietary workgroup to revise the DASH-aligned menus to deliver 40–70% of DASH as described above, and the food costs reduced to a 10% increase. CBN was able to subsidize the funding gap with a grant from a local philanthropic organization (Hearts Foundation) for the duration of the project. Going forward, additional resources and funding will be required to sustain DASH-aligned congregate meals as they require less-processed ingredients and more staff time to prepare. The continuation of SMBP among a large number of enrolled participants shows its acceptability and feasibility among study participants. However, the SMBP intervention was technically challenging, time-consuming for participants, and required extensive staff time and effort. The OMRON 10 blue-tooth enabled devices are optimized for personal use, a feature that created many challenges and extra steps for downloads by the research team, especially after the pandemic prohibited physically interacting with the devices. Most seniors did not have smart phones and were not interested/able to synchronize their devices with the cloud server. In future implementations, we would seek to partner with an existing provider-support or peer-supported SMBP program, both to streamline study procedures, and to facilitate the health data-sharing to clinicians who can modify treatment based on the SMBP measures. An important finding was that Black and female participants were more heavily represented in the group of participants who continued SMBP till the end of the study than in the group who stopped. Since Black participants have a higher risk of hypertension, CVD and lack of healthcare, this finding compels further exploration to design effective and accessible health and nutritional interventions for vulnerable and marginalized communities.

### 4.5. Maintenance

This two-year funded study was not designed to implement and assess long-term sustainability at either the individual or the organizational levels. However, the training offered by RU Bionutrition to CBN’s food services team strengthened CBN congregate meal programming across its four senior centers in the long term, providing kitchen staff with nutritional concepts and specific recipe and ingredient recommendations that promote menus with low-sodium, increased potassium, limited processing, and high nutritional content. The study enhanced CBN’s capacity to examine the impact of meal programs on seniors’ health outcomes and food preferences through the development of new data collection tools.

Our direct engagement and collaboration with DFTA and the funder, ACL, throughout the project and continued dissemination activities will contribute to wider dissemination (see www.CDNetwork.org/CBN-DASH for all project menu and SBPM materials and training videos, accessed on 20 September 2022) and adoption of best practices within the broader network of service providers, which includes over 250 senior centers in New York City. Through collaborative and close introspection of the adaptability of senior meal programs to target specific chronic illnesses, this project served to strengthen best practice for the aging services network and embrace a multi-sectoral research model towards strengthened impact.

Collaboration with agencies such as DFTA, and the NYC Department of Health and Mental Hygiene (DOHMH), is essential not only for policy level change, but also to align standards and operational tools (e.g., Simple Servings) that can prevent unanticipated barriers to implementation and sustainability. For example, the new NYC Mayor’s emphasis on plant-based diets has been integrated recently into the NYC Health + Hospitals ambulatory care network, and this aligns well with implementing the DASH diet in Senior Centers.

## 5. Conclusions

This RE-AIM analysis highlights the utility of study findings and their implications for program implementation. First, program *reach* demonstrates the feasibility of delivering DASH-aligned meals through an existing congregate meals program along with SMBP to reach seniors with high BP in low-income areas of NY.

Despite the interruptions of COVID, which limited the project’s ability to implement the full program as planned, study effectiveness shows a promising reduction in blood pressure and uncontrolled hypertension [24]. Since BP reduction would likely be greater among those with higher blood pressure, the effects on BP and hypertension control are most likely underestimated due to the inclusion of some participants with mild or no hypertension. A follow-up study should include a broad population with adequately powered subgroup analyses to detect heterogeneity of treatment effects. Specifically, the study should include stages 1, 2, and 3 hypertensive patients with different antihypertensive medication regimens in order to assess the intervention’s greatest impact.

To facilitate adoption, the study leveraged the seven-year RU-CDN-CBN partnership to engage two CBN senior centers, secure and share federal funding and collaboratively develop the project, protocol, and interventions. This initial study only targeted two CBN sites and both participated. Future studies would target additional senior centers across multiple different types of organizations in different locations, and more fully explore barriers and facilitators to adoption at the regulatory agency, institutional, and neighborhood levels.

Initial implementation demonstrated the feasibility and acceptability of the DASH and SMBP interventions among study participants. Menu adaptations were informed by client preferences both during planning and implementation. Meal attendance remained consistent, and fidelity (DASH meals as-served versus as-planned) and meal satisfaction were high. DASH concordance data during the initial week revealed discrepancies that were adjusted for greater fidelity, signaling that some time may be needed to test and adjust re-designed menus in this setting and population. Plate waste analysis was labor intensive and not always a reliable measure of client preferences. This method may be valuable for periodic rather than constant monitoring.

Seniors were interested in and able to develop and sustain BP monitoring despite initial lack of familiarity and challenging technology. Future iterations of the study can streamline certain labor-intensive elements. The SMBP data proved extremely useful and important to the study outcomes but were challenging to obtain. In future implementation, we would re-design the workflow and partner with an organization already able to support seniors in this self-monitoring activity.

While this project was not designed to assess long-term impact on either sustainability of DASH meals (organizational level) or BP control (participant level), it laid the groundwork for project maintenance by developing a network and infrastructure to provide intervention services and conduct further exploration to build on study findings. The DASH-aligned menus are DFTA-approved and can be utilized in further testing at additional senior centers. In initial meetings about study findings, the DFTA commissioner has expressed interest in replicating the intervention at other senior centers.

In summary, this study found that one senior center network with multiple delivery sites was able to adopt, implement, and offer a manualized multicomponent evidence-based nutrition and behavior intervention to the seniors they serve at two separate locations. Seniors are receptive to self-management of health and nutritional recommendations when resources are available to facilitate adoption. DASH-aligned congregate meals, combined with a program of education and SMBP, were adopted and implemented by both senior centers and reached the target population of low-income seniors. The intervention was implemented with excellent fidelity to the plan, acceptable to both the staff and clients, and was modestly effective, despite interruption by COVID. Future studies should examine wider dissemination and implementation in a broader population of agencies serving diverse clients in neighborhoods that differ in terms of demographics, socioeconomic characteristics, and access to healthier food options. Furthermore, a more integrated approach would engage the medical care team in ongoing monitoring and follow up of blood pressure control and overall CVD risk, as well as aligning with and leveraging other local programs such as the NYC Department of Health and Mental Hygiene’s “Taking the Pressure Off” and Federal US Department of Agriculture (USDA) programs, including Supplemental Nutrition Assistance Program (SNAP) and Senior Farmers’ Market Nutrition Program (SFMNP), both of which can provide sustainable financial support to food insecure older adults, to aid with their continued purchase and consumption of DASH-compatible groceries.

## Figures and Tables

**Figure 1 nutrients-14-04890-f001:**
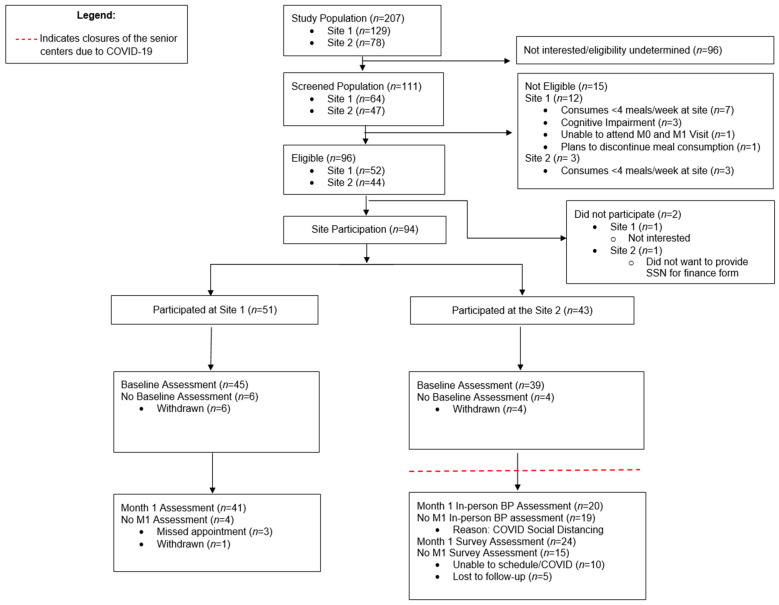
The CONSORT diagram shows the number of participants enrolled at each stage of the study. The closure of each study site due to COVID is also indicated reproduced from [24].

**Figure 2 nutrients-14-04890-f002:**
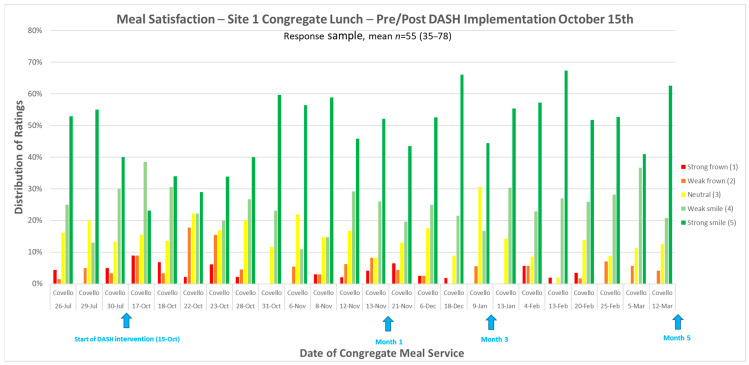
Meal Satisfaction as a measure of acceptability. The colored bars show client responses on Smiley Likert Cards. Meal satisfaction was assessed in July prior to the start of the DASH-aligned menus in October. Blue arrows mark important dates in the intervention- start, month 1, month 3, and month 5.

**Figure 3 nutrients-14-04890-f003:**
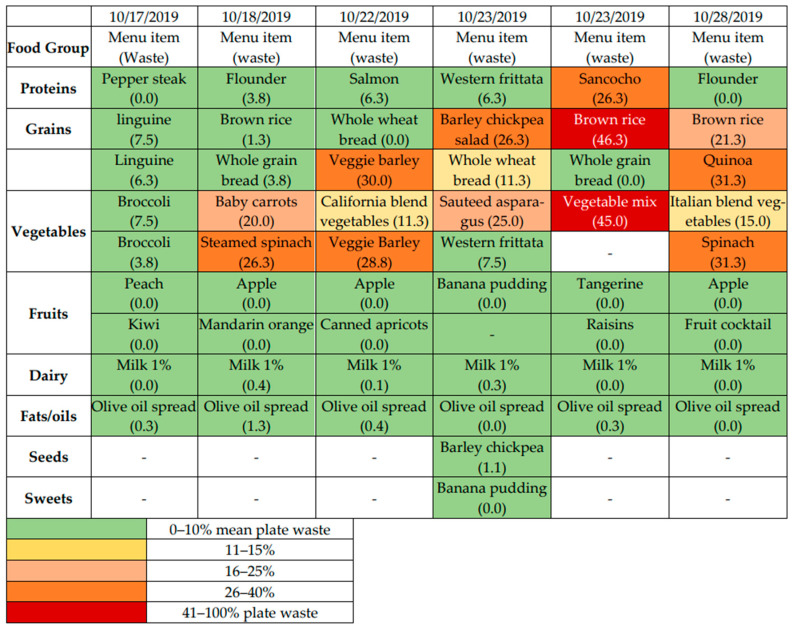
Plate Waste. The average plate waste (% of the component left on the plate) for 20 assessed plates at a given meal is shown for each meal (shown in columns, by date) and broken down into DASH meal components (rows) of proteins, grains, etc. The specific menu item served at each meal is included. The colors represent low (green) versus high (red) levels of plate waste as indicated in the legend above.

**Table 1 nutrients-14-04890-t001:** Study data sources by RE-AIM dimensions.

RE AIM Dimension & Definition [29]	Study Data (Source)
Reach The absolute number, proportion, and representativeness of individuals who are willing to participate in a given initiative, intervention, or program, and reasons why or why not.	Numbers of participants enrolled into study, number of clients served, CONSORT diagram Participant demographics (Baseline surveys) Congregate meal attendance (DFTA tracking in Peer Place)
Effectiveness The impact of an intervention on important individual outcomes, including potential negative effects, and broader impact including quality of life and economic outcomes; and variability across subgroups (generalizability or heterogeneity of effects).	Primary outcome: change in systolic BP at M1 versus Baseline (physiologic measures) Change in BP over time (physiologic measures M3, M6) Change in SMBP (device data download) Improvement in hypertension control rates after intervention Change in Psychosocial and health behavior measures (surveys at Baseline, M1, M3)
Adoption Multiple setting and staff levels: The absolute number, proportion, and representativeness of settings and intervention agents (people who deliver the program) who are willing to initiate a program, and why. Note that adoption can have many (nested) levels; e.g., staff under a supervisor under a clinic or school, under a system, under a community.	Protocol working group meetings (agendas and meeting notes) Advisory committee participation (agendas and meeting notes) DFTA approval of DASH aligned meals (fidelity)
Implementation (Multiple settings and especially delivery staff level): The fidelity to the various elements of an intervention’s key functions or components, including consistency of delivery as intended and the time and cost of the implementation. Importantly, it also includes adaptations made to interventions and implementation strategies and reasons for the above results.	Congregate Meals DASH concordance of as-served, with as-planned (calculated and tracked (feasibility) Client engagement in menu development (townhall attendance and notes) Meal attendance Meal satisfaction (Smiley Likert surveys, plate waste) (acceptability) Economy/efficiency (plate waste, food cost analysis) SMBP Frequency of SMPB measurements (device data downloads) taken at home Data downloads (number sessions, technological hurdles) Training and support burdens (qualitative notes; # sessions training/participant) Barriers and facilitators of SMBP (interaction notes Interaction notes Education Training Attendance (sign-in data BP, nutrition sessions) In-person/web platforms (attendance)
Maintenance (Individual and setting levels): At the setting level, the extent to which a program or policy becomes institutionalized or part of the routine organizational practices and policies. At the individual level, maintenance has been defined as the long-term effects of a program on outcomes after a program is completed. The specific time frame for assessment of maintenance or sustainment varies across projects.	Ability to sustain interventions programmatically ○ Food service staff effort ○ SMBP support ○ Increased food costs Ability to sustain, throughout the study period: ○ SMBP (BP frequency, data downloads) ○ Program (Likert assessments, ○ Ongoing educational support for nutrition, SMBP

**Table 2 nutrients-14-04890-t002:** Study participant and target population demographics.

Demographic Characteristics	Study Participants, Both Sites (*n* = 84)	All Clients, Both Sites (*n* = 291)
Race		
American/Indian/Alaskan Native	1 (1%)	1 (0.3%)
Asian	2 (2%)	58 (20%)
Black	25 (30%)	29 (10%)
Native Hawaiian or Other Pacific Islander	0 (0%)	1 (0.3%)
Multiple Races	4 (5%)	3 (1%)
White	38 (45%)	151 (52%)
Other	5 (6%)	5 (2%)
Unknown or Missing	9 (11%)	43 (15%)
Sex (Female)	57 (68%)	165 (57%)
Age		
60–74	47 (56%)	149 (51%)
75–84	30 (36%)	97 (33%)
85+	6 (7%)	39 (13%)
Unknown	1 (1%)	6 (2%)

**Table 3 nutrients-14-04890-t003:** Concordance of DASH congregate meals as-planned and as-served ^(3)^.

Site 1 (Breakfast and Lunch)				
	As-Planned	As-Served Number of Servings (% of Goal)		
		Week 1	Week 2	Week 3
Protein ^(1)^	15–30	23 (100%)	21 (100%)	23 (100%)
Grains	20	23 (115%)	21 (105%)	18 (90%)
Vegetables	15	16 (107%)	20 (133%)	21 (140%)
Fruit	20	17 (85%)	14 (70%)	19 (95%)
Dairy	10	10 (100%)	10 (100%)	10 (100%)
FAT	10	11 (110%)	7 (70%)	8.5 (85%)
Sweets ^(2)^	4	1 (25%)	1 (25%)	0 (0%)
Nuts, legumes, dried peas and beans	4	3 (75%)	5 (125%)	3 (75%)
**Site 2 (Lunch)**				
	**As-Planned**	**As-Served** **Number of Servings (% of Goal)**		
		**Week 1**	**Week 2**	**Week 3**
Protein ^(1)^	10–20	14 (100%)	15 (100%)	15 (100%)
Grains	10	9 (90%)	10 (100%)	10 (100%)
Vegetables	10	10 (100%)	11 (110%)	11 (110%)
Fruit	10	8 (80%)	11 (110%)	11 (110%)
Dairy	5	5 (100%)	5 (100%)	5 (100%)
FAT	5	6 (120%)	5 (100%)	5 (100%)
Sweets ^(2)^	2–3	0 (0%)	1 (50%)	1 (50%)
Nuts, legumes, dried peas and beans	4	3 (75%)	3 (75%)	3 (75%)

^(1)^ Protein goals were maintained as previously planned at CBN. ^(2)^ Sweets—the goal set was as a maximum, not minimum. ^(3)^ Recommended DASH Diet Servings at 1800 kcals were the minimum servings, if there was a range. Orange cell = over or under the goal, and unacceptable. White cells = met goal or over/under goal, and acceptable.

**Table 4 nutrients-14-04890-t004:** Client food preferences at Research Town Hall.

Excerpt of Responses from Research Town Hall ParticipantsWhat Did You Think of the Samples?		
Site	Sample 1: Spinach	Sample 2: Barley Pilaf
Site 1 *n* = 30	“Very Tasty” “I like the way the spinach was cooked [soft]” “I like to mix both [samples]” 2 participants stated they would not eat either option	“Very Tasty” “I like the onions and carrots. They add flavor. “Where is the meat?”
Site 2 *n* = 53	21 seniors said they liked this sample Positive feedback about the garlic Suggestions were provided for alternative natural flavorings (e.g., nutmeg, lemon, or cinnamon) or accompanying side dishes (e.g., mashed potatoes). Participants pointed out that many took extra samples “Delicious” “Yes!” (regarding the toasted mushrooms) “[I feel like] Popeye baby!” “This was worse [than sample 2]” “I prefer fresh spinach in salad”	24 seniors said they liked this sample “When [barley] is plain there isn’t much flavor, but cooked this way it tasted great” Participants pointed out that many took extra samples “Delicious” Positive and negative feedback about the onions There were some concerns about gluten allergy “Too al dente” “Dry”

**Table 5 nutrients-14-04890-t005:** Barriers and facilitators to self-home blood pressure monitoring.

Participant Reported Barriers and Facilitators during Self-Home Blood Pressure Monitoring	
Barriers	Facilitators
Not at home/traveling (monitor not with them) Forgot to take BP readings Distracted/busy with various activities (e.g., appointment, work, housework, caregiving) Technical problem with BP monitor/cuff (e.g., unpairing, full memory, batteries dying, cuff too tight, error messages) Concern about BP readings (e.g., too high, too low, or incorrect) Challenges/difficulties with study specific procedure (e.g., 2×/day readings, Tru-Read (triplicates), how to take BP reading) General emotional barriers (e.g., being sad, depressed, upset, anxious, worried, stressed, overwhelmed) Sickness/pain due to health issues, illness, or surgery Falls asleep before taking BP; not getting enough sleep, or other sleep issue Issues with specific time periods (e.g., difficult to do AM readings, PM readings, or weekends) Disruptions/lack of routine COVID-19—concern over pandemic, sickness, or caregiving Thought the study ended (due to COVID-19) Concern over home-delivered meals Using another BP monitor (own or someone else’s) BP monitor is heavy, takes up space, or is not portable Not having hypertension (being told by doctor or believing this to be the case)	Visual monitor is kept where participant can see it Routine home BP reading is part of their daily routine; it is usually paired with another activity the participant does daily (e.g., using bathroom, taking medication, eating breakfast) Reminders—different sources of reminders included ACL DASH staff, family, calendar, alarm, and phone. Caring about health (e.g., blood pressure) Caring about research Obligation—sense of duty being in study Familiarity—prior experience with self-home BP monitoring; they previously owned a BP monitor Curiosity about BP readings Having assistance when taking readings

## Data Availability

The data presented in this study are available on request from the corresponding author. The data are not publicly available, specifically, language in the informed consent form addresses the privacy protections if the data were to be viewed by someone not on the study team (deidentified); however, the investigators did not request and the IRB did not approve depositing the data into a public data repository.

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
