# Peer review of "Implementing DASH-Aligned Meals and Self-Measured Blood Pressure to Reduce Hypertension at Senior Centers: A RE-AIM Analysis"

_nutrients, 2022, doi:10.3390/nu14224890_

Round 1

Reviewer 1 Report

This is a single-arm trial that used DASH and SMBP as intervention. My comments are as follows:

1/ introduction
- the knowledge gap is not clear. DASH diet and SMBP are both well-established evidence-based treatment for hypertension. So there is almost no doubt that this would reduce blood pressure in patients with hypertension. The authors did not really point out the knowledge gap or research questions here.
- In my opinion, the knowledge gap is really whether we can successfully implement these interventions in congregate meal settings. So, rather than discussing each of the elements in the introduction, I think the authors should discuss (i) why implementation of these interventions will be difficult /important in this population, and (ii) any other similar studies that implement these interventions in certain population and where the knowledge gap is.

2/ methods:
- for clinical trials, we usually need to register on a clinical trial database before recruiting the first participant
- under study design, the authors seem to state their aim - to test whether DASH and SMBP can lower SBP. However, as the authors also pointed out, both of these interventions have been tested in so many populations and are "evidence-based". Why will these participants be different? Should this be really about the implementation of these evidence-based interventions?
- if SBP is really the primary outcome, then more description on how this is taken is very important. We know that healthcare professionals have poor techniques in measuring BP. How are these taken? Is validated machine used? what is the resting period prior to measurement? how many readings are taken for each visit?
- furthermore, if the primary aim is to detect whether DASH and SMBP can reduce BP, a parallel arm trial is more appropriate
- why include people with normotension if you primary objective is to see whether DASH and SMBP can reduce BP? people with normal BP have limited potential to show BP reduction as their BP is normal already by definition. This has weaken your power to detect any BP changes. Furthermore, these people may be much less motivated to have any lifestyle changes

Results
- the change in SBP is actually not significant (p=0.0713). However, the abstract presented this as if the result is significant. Perhaps we should add p-values in the abstract. Furthermore, is it ethical to present the SBP results again here if this is previously published? (this especially apply to readers who may only have access to the abstract)
- as for the implementation parameters, it is not easy to interpret as there was no pre-determined cut-off. For example, “29/45 (64%) of site 1 participant and 38/39(98%) of site 2 participants attended educational sessions.” Is 64% good enough? Does this mean a success of the educational program? The same observation appeared throughout the whole result section. It is not easy for me as a reader to decide whether this mean the program is implementable. The result section is too long and not easy to read.

Discuss:
- although the authors discussed that the interventions reach the participants well, most patients actually are not interested in the study – they approached 207 participants and finally only 57 retained at 6-month. Does this really mean the program is acceptable to the participants?

Author Response

 - the knowledge gap is not clear. DASH diet and SMBP are both well-established evidence-based treatment for hypertension. So there is almost no doubt that this would reduce blood pressure in patients with hypertension. The authors did not really point out the knowledge gap or research questions here.

- In my opinion, the knowledge gap is really whether we can successfully implement these interventions in congregate meal settings. So, rather than discussing each of the elements in the introduction, I think the authors should discuss (i) why implementation of these interventions will be difficult /important in this population, and (ii) any other similar studies that implement these interventions in certain population and where the knowledge gap is.

We agree with the reviewer that DASH and SMBP are both well-established evidence-based interventions for reducing blood pressure, and the unique focus of this initial study was implementation through a congregate meal program in two settings. The study identified and addressed barriers to implementation related to 1) clients’ dietary preferences, 2) senior centers’ institutional ability to change their food ordering and meal preparation and 3) regulatory oversight and approval by meal funding agencies (NYC Department for the Aging). A review of the literature found very few interventions that adapted the DASH diet to a community setting (Baker et al. 2016), or focused on education about the diet (Ivery et al. 2017), and none that changed the composition of the meals provided at senior centers. Research on SMBP with older adults shows that seniors can benefit from self-monitoring but complex technology presents challenges (Evans et al. 2016). In response to the reviewer’s comment, these knowledge gaps regarding DASH and SMBP implementation in the community-based setting of a senior center, has been further highlighted in the introduction (lines 92-95, 98-100, 105-106).

2/ methods:
- for clinical trials, we usually need to register on a clinical trial database before recruiting the first participant

The trial was registered on clinicaltrials.gov. The CT.gov trial registration number (NCT03993808) has been added to line 121 and is also included in the Institutional Review Board Statement.

- under study design, the authors seem to state their aim - to test whether DASH and SMBP can lower SBP. However, as the authors also pointed out, both of these interventions have been tested in so many populations and are "evidence-based". Why will these participants be different? Should this be really about the implementation of these evidence-based interventions?

Our previous manuscript (Hashemi et al. 2022) presented blood pressure changes with a focus on clinical outcomes data. We agree that this manuscript is focused on the implementation of these two evidence-based interventions in a novel non-healthcare community setting. Accordingly, we updated the section on study design (lines 125-128) in response to the reviewer comment.  

- if SBP is really the primary outcome, then more description on how this is taken is very important. We know that healthcare professionals have poor techniques in measuring BP. How are these taken? Is validated machine used? what is the resting period prior to measurement? how many readings are taken for each visit?

We agree that the method of blood pressure measurement is important because of high variability. Studies such as these conducted in a real-world setting require some pragmatic adaptations to minimize observer and measurement bias. Two of our approaches here directly address this issue: (1) we worked with an independent health contractor who used validated, automated blood pressure devices and standard operating procedures to maintain consistency in measurement, and the home SMBP devices were easy to use, validated automated digital devices with a readout, for which we provided hands-on training in their use. These details have now been added to the manuscript (lines 133-137).

 - furthermore, if the primary aim is to detect whether DASH and SMBP can reduce BP, a parallel arm trial is more appropriate

We agree that a parallel arm (or even a stepped wedge cluster) randomized trial would be desirable, and our future plans include such a study. This pilot study using existing resources was necessary to demonstrate feasibility/acceptability of implementing the two interventions in the senior center/congregate meal setting as well as to measure a signal of effectiveness.

- why include people with normotension if you primary objective is to see whether DASH and SMBP can reduce BP? people with normal BP have limited potential to show BP reduction as their BP is normal already by definition. This has weaken your power to detect any BP changes. Furthermore, these people may be much less motivated to have any lifestyle changes

There were several reasons why we did not exclude senior center participants based on blood pressure. 1) Our previous research with CBN clients (as mentioned in lines 79-80) found that most clients (84%) had elevated BP, and we believed that they could all benefit from the DASH diet. 2) Ordering and preparing multiple sets of meals for on-study and off-study clients was not feasible for senior center staff, and restricting DASH menus to a subset of clients would not be consistent with senior center culture. 3) Segregating clients by their blood pressure status could also be potentially stigmatizing. We agree that this may weaken the power to detect changes, nevertheless we did detect BP changes, making the results especially promising.  

Results
- the change in SBP is actually not significant (p=0.0713). However, the abstract presented this as if the result is significant. Perhaps we should add p-values in the abstract. Furthermore, is it ethical to present the SBP results again here if this is previously published? (this especially apply to readers who may only have access to the abstract)

We have revised the abstract to include the p values. We are including previously published SBP results since efficacy (here, effectiveness) is one of the components of the RE-AIM framework. Previously reported results have been cited appropriately.   

 - as for the implementation parameters, it is not easy to interpret as there was no pre-determined cut-off. For example, “29/45 (64%) of site 1 participant and 38/39(98%) of site 2 participants attended educational sessions.” Is 64% good enough? Does this mean a success of the educational program? The same observation appeared throughout the whole result section. It is not easy for me as a reader to decide whether this mean the program is implementable.

Due to the novel intervention and approach, it is difficult to find similar studies to compare attendance rates. We leave it to the reader to judge the adequacy of our implementation. Nevertheless, the high rate of attendance at site 1 and virtually universal attendance at site 2 suggests very high levels of uptake.

 The result section is too long and not easy to read.

Since we are using the RE-AIM framework, we are presenting results across the multiple specified RE-AIM dimensions (5) and hope the readers will find the detailed information valuable. We have shortened the Results section by moving Table 5, Mission/Services of Advisory Committee Members into the Supplemental materials.

Discuss:
- although the authors discussed that the interventions reach the participants well, most patients actually are not interested in the study – they approached 207 participants and finally only 57 retained at 6-month. Does this really mean the program is acceptable to the participants?

We respectfully disagree that the program was not acceptable.  All 207 senior center clients received the DASH-aligned meals (even if not enrolled in the formal study), and the unchanged meal attendance and high meal satisfaction rates across the clientele after the introduction of the DASH menus reveal high acceptance of the program. Those retained at 6-months were willing to do study surveys and bring in their home BP devices for download by study staff after the interruptions of COVID and does not necessarily signal a lack of acceptance of the program.

It might have been more precise to start the Consort diagram at 111 to reflect the clients who were actually screened for enrollment for the reasons below, however as it has been published in Hashemi et al 2022, we cannot adjust it here.

The invitation to the 207 was informal, not screening, and the responses not tracked in detail. However most of the 96 who did not enter screening fell into these three categories: 1) A subset of clients were Chinese-speaking only, and the validated survey instruments were not available validated in Chinese so those clients could not be considered for enrollment in the formal study. 2) Another group of clients preferred to rotate to different senior centers over the course of a given week to sample different congregate menu and socialize across multiple programs. They were not eligible because they would not meet the congregate meal participation threshold at the DASH intervention site.  

Finally, for some clients, the anticipated time required for completing multiple surveys, and scheduling study visits that were potentially in conflict with highly valued social time, were disincentive to participant, and those clients did not complete screening.  These multiple barriers to enrollment were not specific to the DASH or SMBP interventions but were specific to aspects of the study evaluation measures. Upon analysis, the surveys did not contribute valuable insights, and therefore in future studies, surveys can be eliminated (eliminating the language exclusion for Chinese-speaking and other populations), and extended visits can be significantly reduced.

However, for purposes of equity at the senior center, the DASH aligned meals were provided to all clients, the BP and nutrition education were provided in English, Spanish and Chinese translation to reach all clients, and the meal satisfaction cards were designed with non-textual graphic cues to include all clients in the meal intervention and its evaluation, even though they could not formally participant in the study.

These details were summarized and added to the Results (lines 240-245) and the Discussion (lines 560-566) sections.

Reviewer 2 Report

Reviewer Comments for Manuscript ID: nutrients-2000884

Title: Implementing DASH-Aligned Meals and Self-Measured Blood Pressure to Reduce Hypertension at Senior Centers: A RE-AIM Analysis

Overall impression: Very well-conceived and implemented study halted by COVID-10 pandemic.

My only comment is the readability of the table at the top of page 14. Because the data is critical to the explanation, I would recommend a full page presented on landscape format. For me, the yellow is hard to distinguish.

Otherwise this is an excellent article. I particularly enjoyed the Plate Waste analysis and the frustrations about fidelity among serving sizes.

Author Response

Overall impression: Very well-conceived and implemented study halted by COVID-10 pandemic.

My only comment is the readability of the table at the top of page 14. Because the data is critical to the explanation, I would recommend a full page presented on landscape format. For me, the yellow is hard to distinguish.

Otherwise this is an excellent article. I particularly enjoyed the Plate Waste analysis and the frustrations about fidelity among serving sizes.

We thank the reviewer for this positive feedback. In response to the comment, we have changed the yellow color, enlarged the figure and included it in landscape format to enhance readability of the table.